# ReLU to the Rescue: Improve Your On-Policy Actor-Critic with Positive Advantages

## Abstract

This paper introduces an effective and practical step toward approximate Bayesian inference in on-policy actor-critic deep reinforcement learning. This step manifests as three simple modifications to the Asynchronous Advantage Actor-Critic (A3C) algorithm: (1) applying a ReLU function to advantage estimates, (2) spectral normalization of actor-critic weights, and (3) incorporating *dropout as a Bayesian approximation*. We prove under standard assumptions that restricting policy updates to positive advantages optimizes for value by maximizing a lower bound on the value function plus an additive term. We show that the additive term is bounded proportional to the Lipschitz constant of the value function, which offers theoretical grounding for spectral normalization of critic weights. Finally, our application of dropout corresponds to approximate Bayesian inference over both the actor and critic parameters, which enables prudent *state-aware* exploration around the modes of the actor via Thompson sampling. Extensive empirical evaluations on diverse benchmarks reveal the superior performance of our approach compared to existing on- and off-policy algorithms. We demonstrate significant improvements for median and interquartile mean metrics over PPO, SAC, and TD3 on the MuJoCo continuous control benchmark. Moreover, we see improvement over PPO in the challenging ProcGen generalization benchmark.

## 1 Introduction

Deep Reinforcement Learning (DRL) is a paradigm for finding approximate solutions to complex sequential decision-making problems in domains such as robotics (Ibarz et al., 2021), autonomous driving (Kiran et al., 2021), strategy games (Mnih et al., 2015; Silver et al., 2017; Arulkumaran et al., 2019), and human-computer interaction (Ziegler et al., 2019). In recent years, DRL algorithms have achieved state-of-the-art performance on many challenging benchmarks (Young & Tian, 2019; Lange, 2022; Todorov et al., 2012; Brockman et al., 2016). However, their use in real-world applications depends on their capacity to execute tasks while making policy updates in the face of finite observations of a world in flux. On-policy algorithms, such as Proximal Policy Optimization (PPO) (Schulman et al., 2017) or Asynchronous Advantage Actor-Critic (A3C) (Mnih et al., 2016), update differentiable policies based on recent interactions with the environment. This recency bias and the capacity to actively sample informative observations make on-policy approaches compelling candidates for applications in dynamic real-world environments.

A key facet of active sampling is exploration. On-policy actor-critic methods typically incorporate exploration through entropy regularization or by learning a homogeneous variance parameter for continuous action spaces (Schulman et al., 2015a; Mnih et al., 2016; Schulman et al., 2017). While effective, such on-policy DRL exploration methods are *state-agnostic*, promoting exploration equally regardless of the novelty or familiarity of a given state. This work is motivated by a desire to introduce effective and grounded uncertainty-aware exploration to the on-policy actor-critic framework. Approximate Bayesian inference over actor weights would enable principled exploration, but it is not straightforward due to the policy-gradient objective for optimizing the actor. Thus, we ask, "What is the minimal step we can take toward approximate Bayesian inference," and present VSOP (which stands for Variational [b]ayes, Spectral-normalized, On-Policy reinforcement learning). VSOP consists of three simple modifications to the A3C algorithm: (1) applying a ReLU function to advantage estimates, (2) spectral normalization of actor-critic weights, and (3) incorporating *dropout as a Bayesian approximation* (Gal & Ghahramani, 2016). Under standard

assumptions, we prove that restricting policy updates to positive advantages maximizes value by maximizing a lower bound on the value function plus an additive term. We show that the additive term is bounded proportional to the Lipschitz constant of the value function, grounding the use of spectral normalization as a Lipschitz constant regularizer. Lastly, we show how dropout corresponds to approximate Bayesian inference over the actor and critic parameters, which enables state-aware exploration via Thompson sampling.

Through our thorough empirical assessments on the Gymnasium and Brax MuJoCo benchmarks for continuous control (Brockman et al., 2016; Freeman et al., 2021), we show that VSOP can significantly outperform existing DRL algorithms such as A3C, PPO, SAC, and TD3 for median and interquartile mean (ICM) metrics (Agarwal et al., 2021). Moreover, VSOP significantly outperforms PPO on the challenging ProcGen generalization benchmark, demonstrating improved performance when deployed under distribution shift.

## 2 BACKGROUND

**Notation.** We consider a discounted, T-horizon Markov Decision Process (MDP) defined by the tuple $(\mathcal{S}, \mathcal{A}, \mathrm{P}, \mathrm{r}, \gamma)$, where $\mathcal{S}$ is the state space, $\mathcal{A}$ is the action space, $\mathrm{P}$ is the state transition probability, $\mathrm{r}$ is the immediate reward upon transitioning from state $\mathbf{s}$ to state $\mathbf{s}'$ under action $\mathbf{a}$, and $\gamma \in [0, 1]$ is the discount factor. MDPs provide a framework for modeling sequential decision-making problems, where an agent interacts with an environment over discrete time steps to achieve a goal (Puterman, 2014). Following the notation of Sutton & Barto (2018), we define states at time $\mathrm{t} \in \mathrm{T}$ by the $d$-dimensional, real-valued, random variable, $\mathbf{S}_\mathrm{t} : \Omega \rightarrow \mathcal{S} \subseteq \mathbb{R}^d$, with observable instances $\mathbf{s}_\mathrm{t} = \mathbf{S}_\mathrm{t}(\omega_\mathrm{t}) : \forall \omega_\mathrm{t} \in \Omega$. We define actions by the $m$-dimensional random variable $\mathbf{A}_\mathrm{t} : \Omega \rightarrow \mathcal{A}$, with observable instances, $\mathbf{a}_\mathrm{t} = \mathbf{A}_\mathrm{t}(\omega_\mathrm{t}) : \forall \omega_\mathrm{t} \in \Omega$. Rewards are defined by the continuous-valued random variable, $\mathrm{R}_\mathrm{t} : \Omega \rightarrow \mathcal{R} \subseteq \mathbb{R}$, with observable instances, $\mathrm{r}_\mathrm{t} = \mathrm{R}_\mathrm{t}(\omega_\mathrm{t}) : \forall \omega_\mathrm{t} \in \Omega$. Let the random variable, $\mathrm{G}_\mathrm{t} := \sum_{\mathrm{k=t+1}}^{\mathrm{T}} \gamma^{\mathrm{k}-1-\mathrm{t}} \mathrm{R}_\mathrm{k}$, denote the discounted return. We use the standard definitions for the conditional action distribution/density (policy), $\pi(\mathbf{a} \mid \mathbf{s})$, the state value function under the policy, $v_\pi(\mathbf{s}) := \mathbb{E}_\pi [\mathrm{G}_\mathrm{t} \mid \mathbf{S}_\mathrm{t} = \mathbf{s}]$, and state-action value function under the policy, $q_\pi(\mathbf{s}, \mathbf{a}) := \mathbb{E}_\pi [\mathrm{G}_\mathrm{t} \mid \mathbf{S}_\mathrm{t} = \mathbf{s}, \mathbf{A}_\mathrm{t} = \mathbf{a}]$.

**On-policy Actor-critic reinforcement learning.** On-policy, Actor-critic approaches to reinforcement learning are called *policy-gradient* methods. They directly optimize a policy function, $\pi(\mathbf{a} \mid \mathbf{s}, \boldsymbol{\theta})$, differentiable with respect to parameters, $\boldsymbol{\theta}$, to maximize the expected discounted return under the policy, $v_\pi(\mathbf{s})$. On-policy approaches differ from off-policy approaches in that they only use recent samples from the current policy to achieve this objective. Actor-critic methods differ from other policy-gradient methods because they fit an approximate value function (critic), $v(\mathbf{s}, \mathbf{w})$, to the data collected under the policy, in addition to optimizing the policy function (actor). The critic is typically used in actor optimization but not generally for decision-making.

Deep reinforcement learning implements the actor and critic using neural network architectures, where the function parameters correspond to network weights. We denote the parameters of the actor and critic networks as $\boldsymbol{\theta}$ and $\mathbf{w}$, respectively. The output likelihood of the actor makes distributional assumptions informed by characteristics of the action space, $\mathcal{A}$. A common choice for continuous action spaces is an independent multivariate normally distributed likelihood with homogeneous noise variance, $\pi(\mathbf{a}_\mathrm{t} \mid \mathbf{s}_\mathrm{t}, \boldsymbol{\theta}) \sim \mathcal{N}(\mathbf{a} \mid \boldsymbol{\mu}(\mathbf{s}, \boldsymbol{\theta}), \mathrm{I}\boldsymbol{\sigma}^2(\boldsymbol{\theta}))$, where $\boldsymbol{\sigma}^2(\boldsymbol{\theta}) = (\sigma_1^2, \ldots, \sigma_m^2)$ is the vector of inferred action noise variances. For discrete action spaces, the likelihood is often a categorical distribution, $\pi(\mathbf{a} \mid \mathbf{s}, \boldsymbol{\theta}) \sim \mathrm{Categorical}(\mathbf{a} \mid \boldsymbol{\mu}(\mathbf{s}, \boldsymbol{\theta}))$. In both cases, the mean parameter of the likelihood, $\boldsymbol{\mu}(\mathbf{s}, \boldsymbol{\theta})$, is the $m$-dimensional, vector-valued output of a neural network architecture with parameters, $\boldsymbol{\theta}$. Critic networks are commonly fit using a mean squared error objective, which implies a univariate normally distributed output likelihood with unit variance, $\mathrm{G} \mid \mathbf{s}, \mathbf{w} \sim \mathcal{N}(\mathrm{g} \mid v(\mathbf{s}, \mathbf{w}), 1)$, where the mean parameter is the approximate value function, $v(\mathbf{s}, \mathbf{w})$, and is given by the scalar-valued output of any neural network architecture with parameters, $\mathbf{w}$.

The baseline on-policy, actor-critic policy gradient algorithm performs gradient ascent with respect to the "performance" function, $J(\boldsymbol{\theta}) := v_\pi(\mathbf{s}_0, \boldsymbol{\theta})$, where $v_\pi(\mathbf{s}_0, \boldsymbol{\theta})$ is the value function with respect to the parameters $\boldsymbol{\theta}$. By the policy gradient theorem (Sutton et al., 1999), we have: $\nabla_{\boldsymbol{\theta}} J(\boldsymbol{\theta}) = \nabla_{\boldsymbol{\theta}} v_\pi(\mathbf{s}_0) \propto \int_{\mathcal{S}} \rho(\mathbf{s}) \int_{\mathcal{A}} q_\pi(\mathbf{s}, \mathbf{a}) \nabla_{\boldsymbol{\theta}} \pi(\mathbf{a} \mid \mathbf{s}, \boldsymbol{\theta}) d\mathbf{a} d\mathbf{s}$. Sutton & Barto (2018) show that a generalization of this result includes a comparison of the state-action value function, $q_\pi(\mathbf{s}, \mathbf{a})$, to

an arbitrary baseline that does not vary with the action, $\mathbf{a}$. When the baseline is the state value function, $v_\pi(\mathbf{s})$, we have an objective in terms of the *advantage function* (Schulman et al., 2015b), $h_\pi(\mathbf{s}, \mathbf{a}) \coloneqq q_\pi(\mathbf{s}, \mathbf{a}) - v_\pi(\mathbf{s})$, namely: $\nabla_{\boldsymbol{\theta}} J(\boldsymbol{\theta}) \propto \int_{\mathcal{S}} \rho(\mathbf{s}) \int_{\mathcal{A}} h_\pi(\mathbf{s}, \mathbf{a}) \nabla_{\boldsymbol{\theta}} \pi(\mathbf{a} \mid \mathbf{s}, \boldsymbol{\theta}) d\mathbf{a} d\mathbf{s}$. This formulation in terms of *all actions* can be further simplified in terms of sampled actions and states as $\nabla_{\boldsymbol{\theta}} J(\boldsymbol{\theta}) \propto \mathbb{E}_\pi [h_\pi(\mathbf{S}_t, \mathbf{A}_t) \nabla_{\boldsymbol{\theta}} \log \pi(\mathbf{A}_t \mid \mathbf{S}_t, \boldsymbol{\theta})]$. We use $\mathbb{E}_\pi$ to denote an expectation over states $\mathbf{S}_t$ and actions $\mathbf{A}_t$ collected under the policy $\pi(\mathbf{a} \mid \mathbf{s})$.

In general, because neither the state-action, $q_\pi(\mathbf{s}, \mathbf{a})$, nor the state value, $v_\pi(\mathbf{s})$, functions are known, we need an estimator for the advantage function. For compactness, we will focus on the generalized advantage estimator (GAE) proposed by Schulman et al. (2015b): $h(\mathbf{s}_t, \mathbf{r}_t, \mathbf{w}) = \sum_{k=t+1}^{T} (\gamma\lambda)^{k-1-t} \delta_{t-k+1}^{\mathbf{w}}$, where $0 < \lambda \le 1$, and $\delta_t^{\mathbf{w}} = \mathbf{r}_t + \gamma v(\mathbf{s}_{t+1}; \mathbf{w}) - v(\mathbf{s}_t; \mathbf{w})$ is the temporal difference (TD) residual of the value function with discount, $\gamma$ (Sutton & Barto, 2018). The GAE then yields a low-variance gradient estimator for the policy function: $\widehat{\nabla_{\boldsymbol{\theta}} J}(\boldsymbol{\theta}) \coloneqq \mathbb{E}_\pi [h(\mathbf{S}_t, R_t, \mathbf{w}) \nabla_{\boldsymbol{\theta}} \log \pi(\mathbf{A}_t \mid \mathbf{S}_t, \boldsymbol{\theta})]$. Finally, the actor and critic networks are generally optimized by using mini-batch stochastic gradient descent Robbins & Monro (1951) to fit the functions induced by the network weights to a batch of data collected under the current policy, $\mathcal{D}_\pi^b = \{\mathbf{s}_i, \mathbf{a}_i, r_i\}_{i=1}^b$. The parameter updates are given by,

$$\boldsymbol{\theta} \leftarrow \boldsymbol{\theta} - \eta \frac{1}{b} \sum_{i=1}^{b} h(\mathbf{s}_i, r_i, \mathbf{w}) \nabla_{\boldsymbol{\theta}} \log \pi(\mathbf{a}_i \mid \mathbf{s}_i, \boldsymbol{\theta}), \tag{1a}$$

$$\mathbf{w} \leftarrow \mathbf{w} - \eta \frac{1}{b} \sum_{i=1}^{b} \nabla_{\mathbf{w}} \log p(g(\mathbf{s}_i, r_i, \tilde{\mathbf{w}}) \mid \mathbf{s}_i, \mathbf{w}), \tag{1b}$$

where, $\eta$, is the learning rate, $g(\mathbf{s}_i, r_i, \tilde{\mathbf{w}}) = h(\mathbf{s}_i, r_i, \tilde{\mathbf{w}}) + v(\mathbf{s}_i, \tilde{\mathbf{w}})$, and $\tilde{\mathbf{w}}$ are previous parameters.

## 3 METHODS

This work takes a top-down approach to state-aware exploration for on-policy actor-critic DRL. To employ principled exploration strategies, we would like to have approximate posteriors, $q(\boldsymbol{\Theta} \mid \mathcal{D}_{n-1})$ and $q(\mathbf{W} \mid \mathcal{D}_{n-1})$, for the weights of the actor and critic given data, $\mathcal{D}_{n-1} = \{\mathbf{s}_i, \mathbf{a}_i, r_i\}_{i=1}^{|\mathcal{T}_{n-1}|}$, collected under the policy, $\pi(\mathbf{a} \mid \mathbf{s}, \boldsymbol{\Theta}_{n-1})$, over a set of horizons, $\mathcal{T}_{n-1} = T_1^{n-1} \cup T_2^{n-1} \cup \ldots$. However, fastidiously Bayesian, bottom-up approaches often yield strategies less effective than the state-of-the-art. Leaving debates on evaluation aside, we start from the A3C algorithm and ask, "what minimum changes can we make to get close to an approximate posterior?"

Starting with the critic, $v(\mathbf{s}, \mathbf{w})$, this task seems straightforward because we train the critic with mean squared error loss. Hence, we can use the implied likelihood, $\mathcal{N}(g \mid v(\mathbf{s}, \mathbf{w}), 1)$, and use the *dropout as a Bayesian approximation* (Gal & Ghahramani, 2016) framework to infer an approximate posterior density over critic weights, $q(\mathbf{w} \mid \hat{\mathbf{w}}, p_d)$, where $\hat{\mathbf{w}}$ is the variational parameter for the network weights, and $p_d$ is the dropout rate. We outline the resulting optimization procedure in lines 16-17 of Algorithm 1 for a unit Normal prior over critic weights, $p(\mathbf{w}) = \mathcal{N}(\mathbf{w} \mid 0, \mathbf{I})$.

The inference task is less straightforward for the actor because the A3C objective, $\nabla_{\boldsymbol{\theta}} J(\boldsymbol{\theta}) \propto \mathbb{E}_\pi [h_\pi(\mathbf{S}_t, \mathbf{A}_t) \nabla_{\boldsymbol{\theta}} \log \pi(\mathbf{A}_t \mid \mathbf{S}_t, \boldsymbol{\theta})]$, is not merely maximization of the log-likelihood. Instead, the log-likelihoods, $\log \pi(\mathbf{a} \mid \mathbf{s}, \boldsymbol{\theta})$, are scaled by the advantage function, $h_\pi(\mathbf{s}, \mathbf{a})$. The *dropout as Bayesian approximation* framework estimates the integral over the log evidence lower bound objective using Monte-Carlo integration. For a single sample from the approximate posterior density, $\boldsymbol{\theta} \sim q(\boldsymbol{\theta} \mid \hat{\boldsymbol{\theta}}, p_d)$, the integrand is of the general form:

$$\mathcal{L} = \sum_{i=1}^{|\mathcal{D}|} \log p(\cdot \mid \mathbf{s}_i, \boldsymbol{\theta}) - \mathrm{KL}(q(\boldsymbol{\theta}) \| p(\boldsymbol{\theta})). \tag{2}$$

We make the same prior assumption, $p(\boldsymbol{\theta}) = \mathcal{N}(\boldsymbol{\theta} \mid 0, \mathbf{I})$, for the actor as for the critic, so we only need to focus on the first r.h.s., log-likelihood term. Here we introduce a normal-gamma distribution over the actions r.v., $\mathbf{A}_t$, and a random variable $H_t$:

$$p(\mathbf{A}_t, H_t \mid \mathbf{s}, \boldsymbol{\theta}, \tau, \alpha, \beta) \coloneqq \begin{cases} \mathbf{A}_t \mid H_t, \mathbf{s}, \boldsymbol{\theta}, \tau \sim \mathcal{N}\left(\mathbf{a} \mid \mu(\mathbf{s}, \boldsymbol{\theta}), (\tau H_t)^{-1}\right) \\ H_t \mid \alpha, \beta \sim \mathrm{Gamma}(\alpha, \beta). \end{cases} \tag{3}$$

---

**Algorithm 1** VSOP for Dropout Bayesian Neural Networks

---

**Require:** initial state, $\mathbf{s}'$, environment, $p(\mathbf{s}', \mathrm{r} \mid \mathbf{s}, \mathbf{a})$, rollout buffer, $\mathcal{D}$, initial actor parameters, $\widehat{\boldsymbol{\theta}}$, initial critic parameters, $\widehat{\mathbf{w}}$, dopout rate, $p_d$, learning rate, $\eta$, minibatch size, $b$.

1: **while** true **do**
2:      $\mathcal{D} \leftarrow \emptyset$      ▷ reset rollout buffer
3:      **while** acting **do**      ▷ interact with the environment
4:          $\mathbf{s} \leftarrow \mathbf{s}'$      ▷ update current state
5:          $\boldsymbol{\theta} \sim q(\boldsymbol{\theta} \mid \widehat{\boldsymbol{\theta}}, p_d)$ **if** TS **else** $\boldsymbol{\theta} \leftarrow \widehat{\boldsymbol{\theta}}$      ▷ sample actor params if Thompson sampling (TS)
6:          $\mathbf{a} \sim \pi(\mathbf{a} \mid \mathbf{s}, \boldsymbol{\theta})$      ▷ sample action from policy
7:          $\mathbf{s}', \mathrm{r} \sim p(\mathbf{s}', \mathrm{r} \mid \mathbf{s}, \mathbf{a})$      ▷ sample next state and reward from environment
8:          $\mathcal{D} \leftarrow \mathcal{D} \cup \{(\mathbf{s}, \mathbf{a}, \mathrm{r})\}$      ▷ update rollout buffer
9:      $\mathbf{w}^* \leftarrow \widehat{\mathbf{w}}$      ▷ freeze critic weights for advantage estimates
10:      $\beta \leftarrow (1 - p_d) / (2|\mathcal{D}|)$      ▷ set parameter precision
11:      **while** fitting **do**      ▷ update actor and critic
12:          $\{\mathbf{s}_i, \mathbf{a}_i, \mathrm{r}_i\}_{i=1}^b \sim \mathcal{D}$      ▷ sample minibatch from rollout buffer
13:          $\widetilde{\mathbf{w}} \sim q(\mathbf{w} \mid \mathbf{w}^*, p)$ **if** TS **else** $\widetilde{\mathbf{w}} \leftarrow \mathbf{w}^*$      ▷ sample advantage params if TS
14:          $\boldsymbol{\theta} \sim q(\boldsymbol{\theta} \mid \widehat{\boldsymbol{\theta}}, p_d)$      ▷ sample actor parameters
15:          $\widehat{\boldsymbol{\theta}} \leftarrow \widehat{\boldsymbol{\theta}} - \eta \frac{1}{b} \sum_{i=1}^b h^+(\mathbf{s}_i, \mathrm{r}_i, \widetilde{\mathbf{w}}) \nabla_{\boldsymbol{\theta}} \log \pi(\mathbf{a}_i \mid \mathbf{s}_i, \boldsymbol{\theta}) + 2\beta\boldsymbol{\theta}$      ▷ update actor
16:          $\mathbf{w} \sim q(\mathbf{w} \mid \widehat{\mathbf{w}}, p_d)$      ▷ sample critic parameters
17:          $\widehat{\mathbf{w}} \leftarrow \widehat{\mathbf{w}} - \eta \frac{1}{b} \sum_{i=1}^b \nabla_{\mathbf{w}} \log p(\mathrm{g}(\mathbf{s}_i, \mathrm{r}_i, \widetilde{\mathbf{w}}) \mid \mathbf{s}_i, \mathbf{w}) + 2\beta\mathbf{w}$      ▷ update critic

---

Given a dataset, $\mathcal{D} = \{\mathbf{A}_i, \mathbf{S}_i, \mathrm{H}_i\}_{i=1}^N$, and differentiating the log-likelihood of this distribution with respect to $\boldsymbol{\theta}$, we have:

$$\nabla_{\boldsymbol{\theta}} \log p(\mathbf{A}_\mathrm{t}, \mathrm{H}_\mathrm{t} \mid \mathbf{S}_\mathrm{t}, \boldsymbol{\theta}, \dots)$$
$$= \sum_{i=1}^N \nabla_{\boldsymbol{\theta}} \log \left( \frac{\beta^\alpha \sqrt{\tau}}{\Gamma(\alpha)\sqrt{2\pi}} \mathrm{h}_i^{\alpha - \frac{1}{2}} \exp\left(-\beta\mathrm{h}_i\right) \exp\left(-\frac{1}{2}\mathrm{h}_i\tau(\mathbf{a}_i - \mu(\mathbf{s}_i, \boldsymbol{\theta}))^2\right) \right)$$
$$= -\frac{1}{2} \sum_{i=1}^N \mathrm{h}_i \nabla_{\boldsymbol{\theta}} \tau \left(\mathbf{a}_i - \mu(\mathbf{s}_i, \boldsymbol{\theta})\right)^2 \tag{4}$$
$$= -\frac{1}{2} \sum_{i=1}^N \mathrm{h}_i \nabla_{\boldsymbol{\theta}} \log \pi(\mathbf{a}_\mathrm{t} \mid \mathbf{s}_\mathrm{t}, \boldsymbol{\theta}).$$

Thus, the normal-gamma assumption allows us to recover the A3C optimization form in Equation (1a) while enabling approximate Bayesian inference over the actor parameters. However, the advantage function, $h_\pi(\mathbf{s}, \mathbf{a})$, is not gamma distributed, as it can take on values in the range $(-\infty, 0]$, so we cannot plug it directly into Equation (4). This discrepancy motivates our second, though not strictly valid due to the introduction of zeros, minimal step of passing the advantages through a ReLU function, enabling approximate Bayesian inference over the parameters of the actor to obtain the approximate posterior density of the actor, $q(\boldsymbol{\theta} \mid \widehat{\boldsymbol{\theta}}, p_d)$. We outline the resulting optimization procedure in lines 14-15 of Algorithm 1, where $h^+(\mathbf{s}_i, \mathrm{r}_i, \widetilde{\mathbf{w}}) := \max\left(0, h(\mathbf{s}_i, \mathrm{r}_i, \widetilde{\mathbf{w}})\right)$. Note that the clipped advantages as an actor precision modifier have a very intuitive interpretation. When advantage estimates are low (no evidence of past good actions), the variance of the policy will be high, indicating that the actor should explore more. Conversely, when the advantage estimates are high (evidence of past good actions), the variance of the policy will be low, indicating that the actor should explore less.

**What function does VSOP optimize?** Clipping the advantage estimates to be non-negative has been explored in many policy-gradient contexts (Srinivasan et al., 2018; Oh et al., 2018; Petersen et al., 2019; Ferret et al., 2020). Here, we examine how this augmentation affects the optimization procedure in the context of on-policy actor-critic RL and offer a theoretical hypothesis to ground both our method and the Regret Matching Policy Gradient (RMPG) method of Srinivasan et al. (2018).

**Theorem 3.1.** *Let,* $G_t := \sum_{k=t+1}^{T} \gamma^{k-1-t} R_k$, *denote the discounted return. Let* $q_\pi(\mathbf{s}, \mathbf{a}) = \mathbb{E}_\pi [G_t \mid \mathbf{S}_t = \mathbf{s}, \mathbf{A}_t = \mathbf{a}]$, *denote the state-action value function, and* $v_\pi(\mathbf{s}) = \mathbb{E}_\pi [G_t \mid \mathbf{S}_t = \mathbf{s}]$, *denote the state value function, under policy* $\pi(\mathbf{a} \mid \mathbf{s}, \boldsymbol{\theta})$. *Let* $(x)^+ := \max(0, x)$. *Assume, without loss of generality, that rewards,* $R_t$, *are non-negative. Assume that the gradient of the policy,* $\nabla \pi(\mathbf{a} \mid \mathbf{s}, \boldsymbol{\theta})$, *is a conservative vector field. Then, performing gradient ascent with respect to,*

$$\nabla_{\boldsymbol{\theta}} J(\boldsymbol{\theta}) = \mathbb{E}_\pi \left[ \left( q_\pi(\mathbf{S}_t, \mathbf{A}_t) - v_\pi(\mathbf{S}_t) \right)^+ \nabla_{\boldsymbol{\theta}} \log \pi(\mathbf{A}_t \mid \mathbf{S}_t, \boldsymbol{\theta}) \right], \tag{5}$$

*maximizes a lower-bound,* $v_\pi^*(\mathbf{s})$, *on the state value function,* $v_\pi(\mathbf{s})$, *plus an additive term:*

$$v_\pi^*(\mathbf{s}) \le v_\pi(\mathbf{s}) + C_\pi(\mathbf{s}). \tag{6}$$

*Where,* $C_\pi(\mathbf{s}) = \iint \left( \gamma v_\pi(\mathbf{s}') - v_\pi(\mathbf{s}) \right)^+ d\mathbb{P}(\mathbf{s}' \mid \mathbf{S}_t = \mathbf{s}, \mathbf{A}_t = \mathbf{a}) d\Pi(\mathbf{a} \mid \mathbf{S}_t = \mathbf{s})$, *is the expected, clipped difference in the state value function,* $\gamma v_\pi(\mathbf{s}') - v_\pi(\mathbf{s})$, *over all actions,* $\mathbf{a}$, *and next states,* $\mathbf{s}'$, *under the policy given state,* $\mathbf{s}$. *Here, we use* $\int \ldots d\Pi(\mathbf{a} \mid \mathbf{s})$ *to denote* $\sum_{\mathbf{a}} \ldots \pi(\mathbf{a} \mid \mathbf{s})$ *for discrete action spaces and* $\int \ldots \pi(\mathbf{a} \mid \mathbf{s}) d\mathbf{a}$ *for continuous action spaces. Similarly, we use* $\int \ldots d\mathbb{P}(\mathbf{s}' \mid \mathbf{s}, \mathbf{a})$ *to denote* $\sum_{\mathbf{s}'} \ldots p(\mathbf{s}' \mid \mathbf{s}, \mathbf{a})$ *for discrete state spaces and* $\int \ldots p(\mathbf{s}' \mid \mathbf{s}, \mathbf{a}) d\mathbf{s}'$ *for continuous state spaces. We provide proof in Appendix C.1.*

**Bounding** $C_\pi(\mathbf{s})$. For a $K_\pi$-Lipschitz value function and $\gamma = 1$, the additive term is bounded proportional to the expected absolute difference between states:

$$C_\pi(\mathbf{s}) = \iint \left( v_\pi(\mathbf{s}') - v_\pi(\mathbf{s}) \right)^+ d\mathbb{P}(\mathbf{s}' \mid \mathbf{S}_t = \mathbf{s}, \mathbf{A}_t = \mathbf{a}) d\Pi(\mathbf{a} \mid \mathbf{S}_t = \mathbf{s})$$

$$\le = \frac{1}{2} \iint \left| v_\pi(\mathbf{s}') - v_\pi(\mathbf{s}) \right| d\mathbb{P}(\mathbf{s}' \mid \mathbf{S}_t = \mathbf{s}, \mathbf{A}_t = \mathbf{a}) d\Pi(\mathbf{a} \mid \mathbf{S}_t = \mathbf{s}) \quad Lemma\ C.4$$

$$\le \frac{1}{2} \iint K_\pi \left| \left| \mathbf{s}' - \mathbf{s} \right| \right| d\mathbb{P}(\mathbf{s}' \mid \mathbf{S}_t = \mathbf{s}, \mathbf{A}_t = \mathbf{a}) d\Pi(\mathbf{a} \mid \mathbf{S}_t = \mathbf{s}).$$

This interpretation motivates using spectral normalization (Miyato et al., 2018) of the value function estimator weights, $v(\mathbf{s}, \mathbf{w})$, which regulates the Lipschitz constant, $K_\pi$, of the estimator and can improve off-policy DRL performance (Bjorck et al., 2021; Gogianu et al., 2021). Moreover, this bound is not vacuous for the continuous (nor the discrete) action setting. Under weak assumptions, $f(\mathbf{a}, \mathbf{s}) := \int K_\pi \left| \left| \mathbf{s}' - \mathbf{s} \right| \right| d\mathbb{P}(\mathbf{s}' \mid \mathbf{S}_t = \mathbf{s}, \mathbf{A}_t = \mathbf{a})$, is finite for all $\mathbf{a}$. Therefore, $f^*(\mathbf{s}) = \max_{\mathbf{a}} (\int f(\mathbf{a}, \mathbf{s}) d\Pi(\mathbf{a} \mid \mathbf{S}_t = \mathbf{s}))$, exists and is finite, and $C_\pi(\mathbf{s}) \le \frac{1}{2} f^*(\mathbf{s})$.

We term this method VSOP for Variational [b]ayes, Spectral-normalized, On-Policy reinforcement learning. Algorithm 1 details VSOP for dropout BNNs.

**Comments.** The derivation in Equation (4) assumes access to the policy precision parameter, $\tau = 1/\sigma^2$, and samples of $H_t$. In practice, we fit $\tau$ using maximum likelihood estimation and use clipped GAEs to obtain samples of $H_t$. Moreover, it is only valid for continuous action spaces. We evaluate discrete action spaces below but leave theoretical grounding for future work.

Note that while we show in Equation (4) that approximate Bayesian inference of $\boldsymbol{\theta}$ under an assumed policy that scales actor precision, $\tau$, by clipped advantages, $h^+$, yields an equivalent likelihood objective, we do not implement a policy, $\pi(\mathbf{a} \mid \mathbf{s}, \boldsymbol{\theta})$, that includes this scaling. We leave this exploration to future work as it requires joint inference over $\tau$ and $\boldsymbol{\theta}$ and an appropriate state conditional advantage estimator.

Finally, the conservative vector field assumption of Theorem 3.1 assumes that the actor implements a smooth function. This assumption is often broken in practice as non-smooth ReLU activation functions see use in the baselines we compare to. We leave the investigation of using smooth activation functions to future work.

## 4 RELATED WORKS

VSOP is an on-policy RL algorithm. Table 1 compares the gradient of the performance function, $\nabla J(\boldsymbol{\theta})$, for VSOP with those for relevant on-policy algorithms. We discuss each algorithm below.

Table 1: Comparison of performance functions for on-policy methods

| Method | $\nabla J(\boldsymbol{\theta})$ |
|---|---|
| A3C | $\mathbb{E}_\pi\left[h_\pi(\mathbf{S}_t, \mathbf{A}_t)\nabla\log\pi(\mathbf{A}_t\mid\mathbf{S}_t, \boldsymbol{\theta})\right];\qquad h_\pi(\mathbf{S}_t, \mathbf{A}_t) = q_\pi(\mathbf{S}_t, \mathbf{A}_t) - v_\pi(\mathbf{S}_t)$ |
| **VSOP** | $\mathbb{E}_\pi\left[h_\pi^+(\mathbf{S}_t, \mathbf{A}_t)\nabla\log\pi(\mathbf{A}_t\mid\mathbf{S}_t, \boldsymbol{\theta})\right];\qquad h_\pi^+(\mathbf{S}_t, \mathbf{A}_t) = \max\left(0, h_\pi(\mathbf{S}_t, \mathbf{A}_t)\right)$ |
| RMPG | $\mathbb{E}_\pi\left[\int h_\pi^+(\mathbf{S}_t, \mathbf{a})\nabla d\Pi(\mathbf{a}\mid\mathbf{S}_t, \boldsymbol{\theta})\right]$ |
| TRPO | $\mathbb{E}_\pi\left[h_\pi(\mathbf{S}_t, \mathbf{A}_t)\nabla\rho(\mathbf{S}_t, \mathbf{A}_t, \boldsymbol{\theta})\right];\qquad \rho(\mathbf{S}_t, \mathbf{A}_t, \boldsymbol{\theta}) = \frac{\pi(\mathbf{A}_t\mid\mathbf{S}_t, \boldsymbol{\theta})}{\pi(\mathbf{A}_t\mid\mathbf{S}_t, \boldsymbol{\theta}_{\text{old}})}$ |
| PPO | $\mathbb{E}_\pi\left[\min\left(h_\pi(\mathbf{S}_t, \mathbf{A}_t)\nabla\rho(\mathbf{S}_t, \mathbf{A}_t, \boldsymbol{\theta}), \text{clip}\left(h_\pi(\mathbf{S}_t, \mathbf{A}_t)\nabla\rho(\mathbf{S}_t, \mathbf{A}_t, \boldsymbol{\theta}), 1-\epsilon, 1+\epsilon\right)\right)\right]$ |
| DPO | $\mathbb{E}_\pi\left[\nabla\begin{cases}\left(h_\pi(\rho(\boldsymbol{\theta})-1) - a\tanh(h_\pi(\rho(\boldsymbol{\theta})-1)/a)\right)^+ & h_\pi(\mathbf{S}_t, \mathbf{A}_t) \geq 0 \\ \left(h_\pi\log(\rho(\boldsymbol{\theta})) - b\tanh(h_\pi\log(\rho(\boldsymbol{\theta})/b))\right)^+ & h_\pi(\mathbf{S}_t, \mathbf{A}_t) < 0\end{cases}\right]$ |
| CVaR | $\mathbb{E}_\pi\left[\left(\nu_\alpha - G_t\right)^+\nabla\log\pi(\mathbf{A}_t\mid\mathbf{S}_t, \boldsymbol{\theta})\right];\qquad \nu_\alpha := \alpha\text{-quantile of return, } G_t$ |
| RSPG | $\mathbb{E}_\pi\left[\left(G_t - \nu_\alpha\right)^+\nabla\log\pi(\mathbf{A}_t\mid\mathbf{S}_t, \boldsymbol{\theta})\right];\qquad G_t := \sum_{k=t+1}^T \gamma^{k-1-t}R_k$ |
| EPOpt | $\mathbb{E}_\pi\left[\mathbb{1}\left(G_t \leq \nu_\alpha\right)\nabla J(\theta, \mathbf{S}_t, \mathbf{A}_t)\right];\qquad J(\theta, \mathbf{S}_t, \mathbf{A}_t)$ on-policy perf. function |

**Mirror Learning.** *Proximal Policy Optimization (PPO)* (Schulman et al., 2017), improves upon the baseline policy gradient method by constraining the maximum size of policy updates. PPO employs a clipped surrogate objective function to limit the size of policy updates. PPO simplifies the optimization procedure compared to TRPO (Schulman et al., 2015a), making it more computationally efficient and easier to implement. While PPO constrains policy updates based on the ratio between the new and old policies, VSOP constrains policy updates according to the sign of the estimated advantage function. As such, PPO is an instance of the *mirror learning* framework Kuba et al. (2022), whereas VSOP does not inherit the same theoretical guarantees. Lu et al. (2022) explores the Mirror Learning space by meta-learning a "drift" function. They term their immediate result Learned Policy Optimization (LPO). Through its analysis, they arrive at *Discovered Policy Optimisation (DPO)*, a novel, closed-form RL algorithm.

**Regret Matching Policy Gradient (RMPG).** Srinivasan et al. (2018) present a method inspired by their regret policy gradient (RPG) objective, which maximizes a lower-bound on the advantages: $(h(\mathbf{s}, \mathbf{a}))^+ \leq h(\mathbf{s}, \mathbf{a})$. RPG directly optimizes the policy for an estimator of the advantage lower-bound, denoted as $\nabla_{\boldsymbol{\theta}} J^{\text{RPG}}(\boldsymbol{\theta})$. RMPG, being inspired by RPG, has a different objective, $\nabla_{\boldsymbol{\theta}} J^{\text{RMPG}}(\boldsymbol{\theta})$. In both cases, $q(\mathbf{s}, \mathbf{a}, \mathbf{w})$ is a parametric estimator of the state-action value function, $q_\pi(\mathbf{s}, \mathbf{a})$. RMPG has demonstrated improved sample efficiency and stability in learning compared to standard policy gradient methods. VSOP is closely related to RMPG; however, we provide the missing theoretical foundations to ground RMPG (Appendix C.1), extend RMPG from the *all actions* formulation making it more suitable for continuous control (Appendix C.2), and employ the GAE rather than the state-action value function estimator, $q(\mathbf{s}, \mathbf{a}, \mathbf{w})$.

**Thompson Sampling in Deep Reinforcement Learning.** Thompson sampling has been extensively explored in conventional and Deep Q-Learning (Strens, 2000; Wang et al., 2005; Osband et al., 2016; Moerland et al., 2017; Azizzadenesheli et al., 2018) to improve exploration and sample efficiency. Clements et al. (2019) and Nikolov et al. (2018) propose similar sampling-based exploration strategies for Deep Q-Learning. Jiang et al. (2023) propose a Thompson sampling strategy based on an ensemble of quantile estimators of the state-action value distribution. In the context of *policy gradient* methods, related Upper Confidence Bound (UCB) (Ciosek et al., 2019) and Hamiltonian Monte-Carlo (HMC) (Xu & Fekri, 2022) approaches are proposed for off-policy Soft Actor-Critic (SAC) (Haarnoja et al., 2018), and Henaff et al. (2022) proposes an elliptical episodic reward for general use. Igl et al. (2019) propose Selective Noise Injection using fixed dropout masks to sample policies and then actions, but stop short of formalizing this as Thompson sampling. Similarly for Hausknecht & Wagener (2022). We believe our work is the first to formalize and show the benefit of Thompson sampling for on-policy actor-critic methods.

## 5 EXPERIMENTS

We comprehensively evaluate VSOP against on-policy RL methods across various domains, including continuous and discrete action spaces and diverse dimensionalities in both the action and observation spaces. In Section 5.1, we evaluate VSOP on continuous control tasks using the Gymnasium (Brockman et al., 2016) and Gymnax (Lange, 2022) implementations of MuJoCo (Todorov et al., 2012). In Section 5.2, we assess the capacity of VSOP to learn policies that generalize to unseen environments at test time using the ProcGen benchmark (Cobbe et al., 2020). We use the rliable package to evaluate robust normalized median (Median), interquartile mean (IQM), mean (Mean), optimality gap (OG), and probability of improvement (Prob. Improve) metrics Agarwal et al. (2021). Additional results are provided in Appendix F.

### 5.1 MuJoCo

For this evaluation, we build off of Huang et al. (2022)'s CleanRL package which provides reproducible, user-friendly implementations of state-of-the-art reinforcement learning algorithms using PyTorch (Paszke et al., 2019), Gymnasium (Brockman et al., 2016; Todorov et al., 2012), and Weights & (Biases, 2018). We give full implementation details in Appendix E.1.

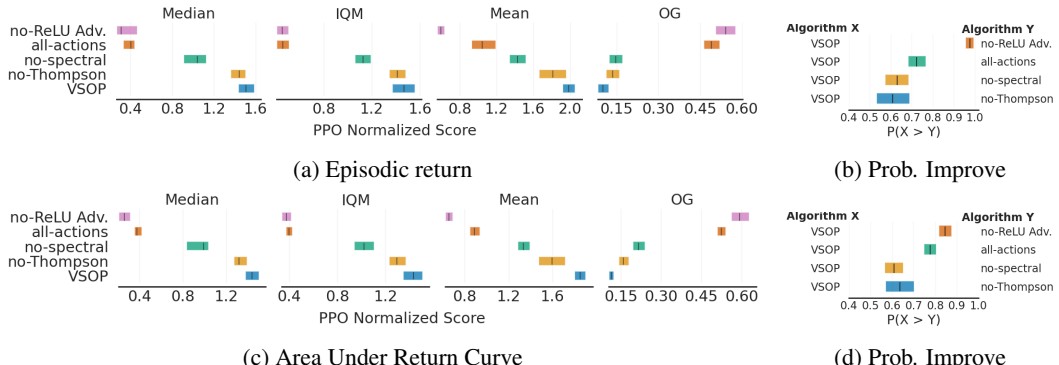

Figure 1: MuJoCo. Ablating the effect of the proposed mechanisms. Here we compare VSOP to VSOP without spectral normalization (no-spectral), VSOP without Thompson sampling (no-hompson), VSOP without advantage clipping (no-ReLU Adv.), and VSOP using all-actions policy optimization (all actions). We see that no single mechanism contributes greater than the sum of all changes, lending credence to the validity of our theory. These results hold for overall performance (a-b), and sample efficiency (c-d). Metrics are computed wrt to the average episodic return of the last 100 episodes and the area under the episodic return curve over 10 random seeds

**Ablation of mechanisms.** First, we investigate the influence of our four proposed mechanisms on the performance of VSOP. For reference, the mechanisms are positive advantages, single-action setting, spectral normalization, and Thompson sampling. To ablate each mechanism, we compare VSOP to four variants: VSOP without advantage clipping (no-ReLU Adv.), VSOP in the all-actions setting (all-actions), VSOP without spectral normalization (no-spectral), and VSOP without Thompson sampling (no-Thompson). We hyperparameter tune each variant in accordance with the same procedure used for VSOP (see Table 2 for details). Figure 1 summarizes these results, and we see clearly that no single mechanism contributes greater than the sum of all changes, lending credence to our theoretical analysis. We see that positive advantages and operating in the single-action regime impact performance on MuJoCo significantly. Spectral normalization and Thompson sampling also influence performance on MuJoCo positively, especially in high-dimensional action and observation space settings such as Humanoid, Humanoid Stand-Up, and Ant, as shown in Figure 8 of Appendix F.2. The performance gains for spectral normalization align with results given by Bjorck et al. (2021) and Gogianu et al. (2021) for DDPG (Lillicrap et al., 2015), DRQ (Kostrikov et al., 2020), Dreamer (Hafner et al., 2019), DQN (Wang et al., 2016) and C51 (Bellemare et al., 2017).

**Comparison to baselines.** Next, we compare VSOP to baseline implementations: PPO, A3C, SAC, and TD3. We use the CleanRL (Huang et al., 2022) implementation of PPO, SAC and TD3; the

StableBaselines3 (Raffin et al., 2021) hyper-parameter settings for A3C. We also include comparisons to RMPG (adapted for continuous control) and VSPPO (PPO with spectral normalization, and Thompson sampling via dropout). We tune RMPG and VSPPO using the same Bayesian optimization (Snoek et al., 2012) protocol as VSOP. Figure 2 summarizes our results, where we see that VSOP shows significant improvement over each baseline with respect to each metric, except for the SAC and TD3's mean scores. See Figure 9 in Appendix F.3 for training curves of these results.

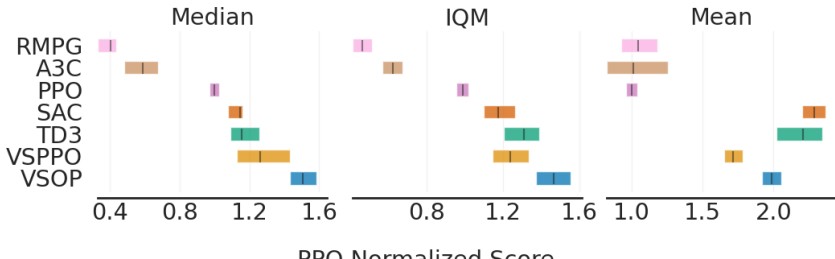

Figure 2: MuJoCo. Comparison to baselines. We see that VSOP (blue) shows significant improvement over each baseline with respect to the Median and IQM metrics. VSOP only trails SAC and TD3 for the mean metric. Metrics are computed wrt to the average episodic return of the last 100 episodes over 10 random seeds

**Effect of asynchronous parallelization.** Following Lu et al. (2022), we also evaluate VSOP on the Brax implementation of MuJoCo in a massively parallel setting. Where in the above experiments we set the number of asynchronous threads to 1 and the number of steps per rollout to 2048, here we set the number of asynchronous threads to 2048 and the number of steps to 10. We see in Figure 3 that while VSOP still outperforms A3C significantly, it trails PPO. Full training curves are shown in Figure 13 of Appendix F.5.

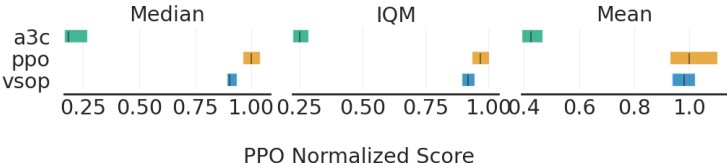

Figure 3: MuJoCo. Comparison to on-policy baselines with extreme parallelization. Here we compare VSOP to on-policy baselines on MuJoCo with 2048 threads and 10 steps per rollout. Metrics are computed wrt to the average episodic return of the last 100 episodes over 20 random seeds

Interestingly, hyper-parameter tuning showed spectral normalization to be detrimental to the performance of VSOP in this massively parallel setting. We investigate the effect of parallelization on VSOP effectiveness and efficiency in Figure 4. Here we set the rollout size to 2048 environment interactions and sweep the number of threads and number of steps. For each configuration, we do a hyper-parameter sweep in MuJoCo Brax using the reacher, hoppper, and humanoid evironments over 1 million environment interactions. We then evaluate on 10 MuJoCo environments over 3 million environment interactions. The blue bars show metrics for VSOP with spectral normalization. We see that VSOP is most effective and efficient with spectral normalization with a low thread count and that for a fixed rollout size, these measures fall with increasing parallelization. For VSOP without spectral normalization, the trend is less clear, but appears to be generally the opposite for a fixed rollout size. This indicates that spectral normalization will be beneficial in applications where it is not feasible to run many parallel agents.

## 5.2 PROCGEN

In lieu of finding a suitable benchmark for continuous control, we assess the capacity of VSOP to generalize to unseen environments using ProcGen (Cobbe et al., 2020). ProcGen is a set of 16 environments where game levels are procedurally generated, creating a virually unlimited set

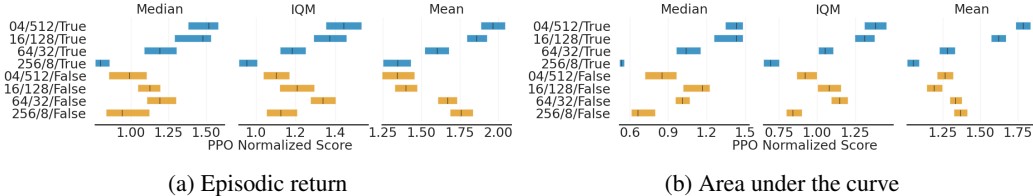

(a) Episodic return

(b) Area under the curve

Figure 4: MuJoCo: effect of parallelization on VSOP. Naming convention: #threads/#steps/spectral norm. We see that VSOP is most effective 4a and most efficient 4b in lower thread settings for a fixed rollout size of 2048 steps when using spectral normalization. Metrics are computed wrt to the average episodic return or area under the curve for the last 100 episodes over 5 random seeds

of unique levels. We follow the "easy" generalization protocol where, for a given environment, models are trained on 200 levels for 25 million time steps and evaluated on the full distribution of environments. We use the same architecture as PPO in the CleanRL library, and do a Bayesian optimization hyper-parameter search using the bossfight environment. We search over the learning rate, GAE $\lambda$, number of minibatches per epoch, number of epochs per rollout, the dropout rate, and the entropy regularization coefficient. Full implementation details are given in Appendix E.3. Figure 5 summarizes our results. We see broad significant improvement over PPO across both the PPO and Min-Max normalized metrics. Furthermore, we see improvement over EDE Jiang et al. (2023) with respect to the IQM, mean, and optimality gap metrics. These results present strong evidence for the suitability of VSOP for deployment in non-stationary environments. See Figure 7 in of Appendix F.1 for full training and test curves.

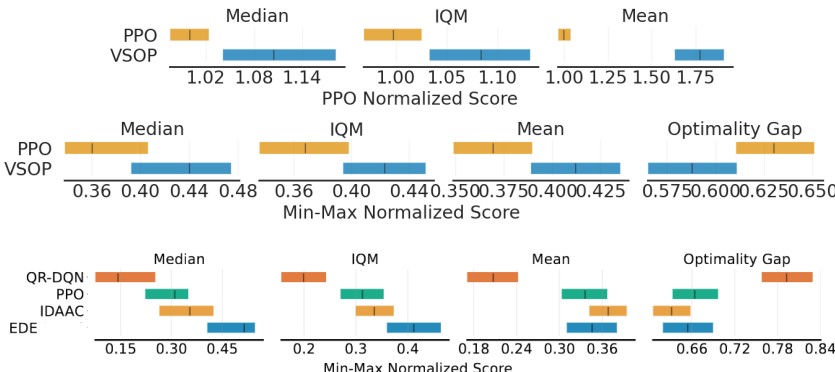

Figure 5: ProcGen comparison to PPO. In the top pane, we see significant improvement over PPO with respect to all metrics for the PPO normalized scores. In the middle pane, we see significant improvement over PPO in terms of the IQM, mean and optimality gap metrics for the Min-Max normalized scores. In the bottom pane, we include results reported by Jiang et al. (2023). It appears as though we improve over EDE with respect to the IQM, mean, and optimality gap metrics. Metrics are computed wrt to the average episodic return of the last 100 episodes over 5 random seeds

## 6 CONCLUSION

This work represents a step towards principled approximate Bayesian inference in the on-policy actor-critic setting. Our method is realized through simple modifications to the A3C algorithm, optimizes a lower bound on value plus an additive term and integrates exploration via Thompson sampling. Our empirical evaluations across several diverse benchmarks confirm our approach's improved performance compared to existing on-policy algorithms.

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
