# OpenReview forum: "ReLU to the Rescue: Improve Your On-Policy Actor-Critic with Positive Advantages"
_ICLR.cc/2024/Conference — Submitted to ICLR 2024_

### Official Review · Reviewer_kpwW · 2023-10-27

**Soundness:** 2 fair
**Presentation:** 1 poor
**Contribution:** 2 fair
**Rating:** 5
**Confidence:** 3

**Summary:**

This paper made three simple modification to advantage actor-critic methods.
(1) introduced a ReLU function to restrict policy update to the optimal policy while enable approximate Bayesian inference.
(2) used spectral normalization to restrict the output of network
(3) used Thopson sampling to do exploration via dropout.
The reported results indicate that, mostly, the proposed method achieves improved returns when compared to the popular on-policy algorithms and other off-policy baseline methods.

**Strengths:**

Minimal modification of the ac method to enable Bayesian inference is an interesting and valuable idea. However, similar discussions exist in previous work[1].

Use Thompson sampling to replace passive exploration without introducing complex machanism and high computational cost is important for dealing with the Exploration-exploitation dilemma for on-policy algorithms.


The performance of this method is impressive.

[1] Levine, Sergey. "Reinforcement learning and control as probabilistic inference: Tutorial and review." arXiv preprint arXiv:1805.00909 (2018).

**Weaknesses:**

This paper is not well written. Neccessary background and discribtion of Thompson sampling is missing. The names of metrics shown in Figure 1(Median, IQM, Mean, OG) should be emphasized above.(e.g. change 'robust normalized median, interquartile mean, and mean metrics' to 'robust normalized median(Median), interquartile mean(IQM), mean(Mean) and optimal gap(OG) metircs'). The multiple use of some terms in many places made me confused, e.g. I think A3C represent two different algorithms respectively in Figure 1 and 2.

Except in the case of sparse rewards, it is generally not acceptable to assume that the difference between the expected value of the value function over the next states and the value function at the current state is zero.

According to the equation (3), the paper assume that $h$ is independent from $\theta$. However, the advantage function is strong depend on current policy and also depend on $\theta$. And the advantage function may not be approximated via Gamma distribution. From my understanding, the $\sigma^2\geq 0$ is a more direct reason why $h$ is a non-negative value.


typo on (4): $\frac{\beta^\alpha\sqrt{\tau}}{\sqrt{\Gamma(\alpha)2\pi}}\rightarrow \frac{\beta^\alpha \sqrt{\tau}}{\Gamma(\alpha)\sqrt{2\pi}}$, $exp(\beta h_i)\rightarrow exp(-\beta h_i)$.

typo in appendix, an extra $\nabla$ after 'Letting, $v^*_\pi(s_0)$......'

**Questions:**

Can you explain in detail how to combine Thompson sampling with Bayesian inference? And why this is a better **state-aware** exploration method?

Is it necessary to assume it as a gamma-normal distribution?

---

> ### Author Response · Authors · 2023-11-18
> **Response to Reviewer kpwW**
>
> We sincerely appreciate your close reading of our paper! We are encouraged that you found the idea of our contribution interesting and valuable, that you recognized low computational overhead of our solution, and that your were impressed by our method's performance. The reviewer brings up some good and challenging points that we would like to address.
>
> ### Questions
>
> > Is it necessary to assume it as a gamma-normal distribution?
>
> The gamma-normal assumption allows us to interpret adding dropout and weight-decay regularization as sensible approximate Bayesian inference without adding complex computational overhead to the original A3C optimization algorithm. As such, this assumption primarily serves to ground Thompson sampling through approximate Bayesian inference and is not requisite for Theorem 1. As with the original result of the policy gradient theorem, the results in Equations (5-6) do not make any distributional assumptions on $\pi$ and should hold for all policies with differentiable probability densities/distributions.
>
> ### Methodoligical Concerns
>
> > Except in the case of sparse rewards, it is generally not acceptable to assume that the difference between the expected value of the value function over the next states and the value function at the current state is zero.
>
> Thank you for this comment as your attention has led to a clear improvement. We hope that our comment to all reviewers above has addressed this concern. Specifically, upon a close re-examination of this assumption, we agree it would not generally hold. Fortunately, we can show that it is not necessary and that $C_\pi(\mathbf{s})$ can be smaller than under the zero-expectation assumption, resulting in a smaller additive:
>
> **Lemma.**
>
> $$\text{ReLU}(a) <= |a|$$
>
> *Proof*.
> $$ \text{ReLU}(a) = \max(0, a)$$
> $$\quad\quad\quad\quad= \frac{1}{2}a + \frac{1}{2}|a|$$
> $$\quad\quad\quad\quad= \\{a  \text{ if } a\geq0\text{ else }0\text{ if } a<0$$
> $$\quad\quad\quad\quad\leq \\{a  \text{ if } a\geq0\text{ else }-a\text{ if } a<0 (-a > 0)$$
> $$\quad\quad\quad\quad= |a| $$
>
> Therefore,
>
> $$C_\pi(\mathbf{s}) = \iint \left(v_\pi(\mathbf{s'}) - v^\pi(\mathbf{s}) \right)^+ dP(\mathbf{s}' \mid \mathbf{S}^\mathrm{t} = \mathbf{s}, \mathbf{A}^\mathrm{t} = \mathbf{a}) d\Pi(\mathbf{a} \mid \mathbf{S}^\mathrm{t} = \mathbf{s})$$
> $$\quad\quad\quad\leq \iint \left|v_\pi(\mathbf{s'}) - v_\pi(\mathbf{s}) \right| dP(\mathbf{s}' \mid \mathbf{S}^\mathrm{t} = \mathbf{s}, \mathbf{A}^\mathrm{t} = \mathbf{a}) d\Pi(\mathbf{a} \mid \mathbf{S}^\mathrm{t} = \mathbf{s}) $$
> $$\quad\quad\quad\leq \iint K_\pi\left|\left|\mathbf{s'} - \mathbf{s} \right|\right| dP(\mathbf{s}' \mid \mathbf{S}^t = \mathbf{s}, \mathbf{A}^\mathrm{t} = \mathbf{a}) d\Pi(\mathbf{a} \mid \mathbf{S}^\mathrm{t} = \mathbf{s})$$
>
>
> Please let us know if you have further concerns about this point.

---

> ### Author Response · Authors · 2023-11-18
> **Presentation Concerns**
>
> ### Presentation Concerns
>
> > Neccessary background and discribtion of Thompson sampling is missing. Can you explain in detail how to combine Thompson sampling with Bayesian inference? And why this is a better state-aware exploration method?
>
> Approximate Bayesian inference over the parameters of the policy, $\theta$, yields a distribution over those parameters, $p(\theta \mid \mathcal{D})$. Sampling a policy from this distribution, $\hat{\theta} \sim p(\theta \mid \mathcal{D})$, is as easy as sampling a dropout mask and then running a forward pass of the network, yielding the likelihood, $\pi(\mathbf{a} \mid \mathbf{s}, \hat{\theta})$. Then sampling an action is done by sampling an action from the sampled policy, $\mathbf{a} \sim \pi(\mathbf{a} \mid \mathbf{s}, \hat{\theta})$. This is precisely the procedure descibed by Thompson sampling. We outline this procedure in lines 5-6 of Algorithm 1 and have a section on Thomspon sampling for RL in the related works section, but we agree with your assessment that this can be made more explicit in the main text and we are working on incorporating the above description to the manuscript. We would appreciate any feedback on it should you have time.
>
> We hypothesize that this is a better state-aware exploration method for two reasons. First, for less frequently visited states the diversity of the sampled paramters of the policy will be greater promoting more exploration. As a state is visited more often under actions that yield positive advantages, the diversity of samples will concentrate promoting less exploration. Thus, we get more exploration for states that we have less experience of good actions, and less exploration in states where we know what actions lead to good expected returns. Second, this exploration is done around the mode of the policy distribution, so the model is less likely to explore actions that are far from the mode, which could be more likely to lead to failure.
>
>
> > The names of metrics shown in Figure 1(Median, IQM, Mean, OG) should be emphasized above.(e.g. change 'robust normalized median, interquartile mean, and mean metrics' to 'robust normalized median(Median), interquartile mean(IQM), mean(Mean) and optimal gap(OG) metircs').
>
> We are adding your suggestions on the metric names.
>
> > The multiple use of some terms in many places made me confused, e.g. I think A3C represent two different algorithms respectively in Figure 1 and 2.
>
> We agree that we can be more clear here and propose making the following changes to the names in Figure 1 in order to help aleviate the confusion:
>
> RMPG -> VSOP "all-actions"
> A3C -> VSOP no-ReLU-Adv.
> No Spectral -> VSOP no-spectral
> No Thompson -> VSOP no-Thompson
>
> > typo['s] on ..
>
> Nice catches. We have updated the manuscript with, $\frac{\beta^{\alpha}\sqrt{\tau}}{\Gamma(\alpha)\sqrt{2\pi}}$ and 'Letting $\nabla v_\pi^*(\mathbf{s}_0)$'.

---

> ### Author Response · Authors · 2023-11-22
> **Thank you, again**
>
> Dear Reviewer,
>
> Thank you again for sharing your valuable feedback with us. Your comments have clearly led to an improved manuscript! We strongly believe that the motivation, theory, and empirical results make this submission worthy of acceptance. Moreover, we think our response addresses your concerns and have updated the paper accordingly. Specifically, we have updated the $C_\pi(\mathbf{s})$ bound, included discussions on the normal-gamma assumption and Thompson sampling, and incorporated your formatting instructions. We also respectfully contend that our approach differs significantly from that of Sergey Levine's in the work you cite. Their max-entropy interpretation has optimal trajectories as the target of probabilistic inference rather than model weights (or functions induced by those weights). The practical implications of this difference become clear in comparing our objective function in Equation (5) of our work to the objective function in section 4.1 of their work, which is precisely the A3C objective. If you have no further concerns, we kindly ask you to consider raising your score. Otherwise, we would also be happy to discuss further, should you find the time.
>
> Best regards,
> The Authors

---

> > ### Comment · Reviewer_kpwW · 2023-11-22
> > **Thanks for reply**
> >
> > Thank you for your comprehensive response, which addressed most of my concerns. I will raise my rating to 5. However, I cannot assign a higher score because the method lacks the novelty required, and when compared to other exploration techniques, it fails to convince me that it is a better state-aware exploration method.

---

> > > ### Author Response · Authors · 2023-11-22
> > > **Score Increase**
> > >
> > > We greatly appreciate the 2-point score increase and are pleased you found our response comprehensive!
> > >
> > > We understand your concerns about novelty, but we would like to highlight that the result in Theorem 3.1 is novel and contributes a missing theory to RMPG. Moreover, our explication of the assumptions that make approximate Bayesian inference of the actor parameters sensible is also a new contribution.
> > >
> > > We have shown that this step towards grounded approximate Bayesian inference for on-policy actor-critic methods improves both sample efficiency and performance in the continuous control setting with respect to baseline in the ablation study (VSOP vs. VSOP no-Thompson). Moreover, we have shown evidence of improvement over the recently published EDE method [1] (specifically designed to investigate the role of exploration for generalization) in the Interquartile Mean, Mean, and Optimality Gap metrics on the ProcGen generalization task. We are also interested in further heads-up comparisons to other state-aware exploration methods but ask you to consider the significant computational investment this requires. Establishing our significant empirical evidence thus far (multiple settings, multiple environments, multiple random seeds, hyperparameter tuning) has already cost over 80,000 hours of GPU time according to Weights & Biases.
> > >
> > > Among these reasons, we believe our presented work is valuable to the reinforcement learning community and deserves acceptance to ICLR 2024.
> > >
> > > Thank you again for your consideration, feedback, and already generous score increase.
> > >
> > > [1] Jiang, Yiding, J. Zico Kolter, and Roberta Raileanu. “On the Importance of Exploration for Generalization in Reinforcement Learning.” NeurIPS (2023).

---

### Official Review · Reviewer_tvnj · 2023-11-01

**Soundness:** 3 good
**Presentation:** 4 excellent
**Contribution:** 3 good
**Rating:** 6
**Confidence:** 3

**Summary:**

The paper presents a modification to asynchronous advantage actor-critic (A3C) that involves incorporating a ReLU function to the advantage estimates, using spectral normalization and incorporating dropout. The key idea in their work is that exploration is required in on-policy reinforcement learning. When there is no exploration, certain states may fail to get explored and the policy might get trapped. In order to provide a remedy to this, algorithms typically use methods that do not depend on the frequency with which states are visited, which can provide suboptimal results compared to using simply using a method that incorporates details of frequency with which states are visited. As a remedy to this, the algorithm incorporates using a ReLU function to the advantage function. The way this is performed is that in the critic step, dropout is employed and in the actor step, ReLU is used. The motivation behind using the ReLU is that it enables a Bayesian inference over the actor's parameters. The rationale for the changes are justified in the Methods section. The work incorporates a theoretical bound that illustrates how their methods allows a maximization of estimation of state value functions plus a constant. The constant is then massaged in the spectral norm refinement stage of the algorithm. There are also very extensive empirical studies.

**Strengths:**

The algorithm provides a remedy to an issue with asynchronous advantage actor-critic algorithms (or, more broadly, reinforcement learning algorithms that many algorithms do not take into account state visitation frequency. This issue has been noted by other works as a very important topic in theoretical RL. See the works of [@book{sutton2018reinforcement, title={Reinforcement learning: An introduction}, author={Sutton, Richard S and Barto, Andrew G}, year={2018}, publisher={MIT press} }, @article{tsitsiklis2002convergence, title={On the convergence of optimistic policy iteration}, author={Tsitsiklis, John N}, journal={Journal of Machine Learning Research}, volume={3}, number={Jul}, pages={59--72}, year={2002} }, @inproceedings{winnicki2023convergence, title={On The Convergence Of Policy Iteration-Based Reinforcement Learning With Monte Carlo Policy Evaluation}, author={Winnicki, Anna and Srikant, R}, booktitle={International Conference on Artificial Intelligence and Statistics}, pages={9852--9878}, year={2023}, organization={PMLR} }] for more on this. The work takes an interesting angle by looking at statistical techniques for improvement which in turn motivated other improvement to the algorithm. The work provides a theoretical intuition and bound as well as numerous empirical studies.

**Weaknesses:**

In the theoretical component of the algorithm, while there is a theoretical result which shows how the value function improves as a result of the modifications, which is very nice, but I think that the work could shed light on the role of these parameters on the overall convergence of the modified A3C? I also noticed that neither the simulations nor the theoretical results shed light on the exact role of the choice of dropout, relu etc., on the impact of their bounds, both theoretical bounds and empirical bounds, with the exception of the justification of the spectral normalization step. Or perhaps, a comparison to A3C, since that is the algorithm the current paper is based on?

**Questions:**

A question I have is whether the constant K is policy dependent, in which case, what would the policy improvement step be optimizing over in (6)? I noticed that the K-Lipschitz assumption is introduced with respect to a particular value function over all $s\in\scriptS,$
which makes me wonder if K is dependent on policies π. Another question I have is if the assumptions on the action distribution follow the normal-gamma in the bound on the value function improvement in (6)? What other assumptions are incorporated in the bound on the value function improvement?

---

> ### Author Response · Authors · 2023-11-18
> **Response to Reviewer tvnj**
>
> # Response to Reviewer tvnj
>
> We are keen to thank the reviewer for their thorough and helpful review. We're glad that the reviewer finds that our work tackles an important topic and that our empirical results are extensive. Thank you for the insightful pointers on this topic, which we are incorporating into the updated manuscript! The reviewer raises many good questions that we would like to answer below.
>
> ### Questions
>
> > I noticed that the K-Lipschitz assumption is introduced with respect to a particular value function over all which makes me wonder if K is dependent on policies π. A question I have is whether the constant K is policy dependent, in which case, what would the policy improvement step be optimizing over in (6)?
>
> This is an excellent question and we are very thankful for your insight. We hope that the response to all reviewers above serves to answer it. In summary, we agree that K is policy dependent in general, and the dependence has given new insights into the role of spectral normalization as a regularizer of K. Please do not hesitate to follow up if you have furhter clarifying questions regarding this point.
>
> > Another question I have is if the assumptions on the action distribution follow the normal-gamma in the bound on the value function improvement in (6)?
>
> As with the original result of the policy gradient theorem, the result in Equation (6) does not make any distributional assumptions on either $q_\pi$, $v_\pi$, or $\pi$, and should hold for all policy distributions with differentiable densities.
>
> > What other assumptions are incorporated in the bound on the value function improvement?
>
> As stated in Theorem 3.1, two assumptions are necessary to get the result in Equation (6): 1.) that rewards, $\mathrm{R}_t$, are non-negative (which we make without loss of generality); and 2.) that the gradient of the policy is a conservative vector field. With the additional assumption that the value function is K-Lipschitz, we can bound the constant as in Equation (7). Note that in response to Reviewer kpwW's comment, we can drop the additional assumption that the expected difference in the value function between subsequent states is 0.
>
> ### Concerns
>
> > In the theoretical component of the algorithm, while there is a theoretical result which shows how the value function improves as a result of the modifications, which is very nice, but I think that the work could shed light on the role of these parameters on the overall convergence of the modified A3C?
>
> We are very interested in establishing convergence rates for our algorithm, but believe that such an analysis would consitute a self contained journal paper akin to the works of *Agarwal et al. 2021* or *Shen et al. 2023*. Note that these works were published several years after the acceptance of the original A3C paperto ICML in 2016.
>
> Agarwal, Alekh, et al. "On the theory of policy gradient methods: Optimality, approximation, and distribution shift." The Journal of Machine Learning Research 22.1 (2021): 4431-4506.
>
> Shen, Han, et al. "Towards Understanding Asynchronous Advantage Actor-critic: Convergence and Linear Speedup." IEEE Transactions on Signal Processing (2023).
>
> > I also noticed that neither the simulations nor the theoretical results shed light on the exact role of the choice of dropout, relu etc., on the impact of their bounds, both theoretical bounds and empirical bounds, with the exception of the justification of the spectral normalization step.  Or perhaps, a comparison to A3C, since that is the algorithm the current paper is based on?
>
> While dropout does not factor into the bound in Equation (6), we can improve the clarity of Theorem 3.1 to emphasize that the bound is directly dependent on applying the ReLU function to the advantage estimates. In Theorem 3.1 we write, $(x)^+ := \max(0, x)$, which is the definition of the ReLU function.

---

> > ### Author Response · Authors · 2023-11-22
> > **Thank you, again**
> >
> > Dear Reviewer,
> >
> > Thank you again for sharing your valuable feedback with us. We strongly believe that the motivation, theory, and empirical results make this submission worthy of acceptance. Moreover, we think our response addresses your concerns and have updated the paper accordingly. Specifically, we updated the implications of the policy on the Lipschitz constant, included a discussion on the normal-gamma assumption, and clarified our assumptions. If you have no further concerns, we kindly ask you to consider raising your score. Otherwise, we would also be happy to discuss further, should you find the time.
> >
> > Best regards,
> > The Authors

---

### Official Review · Reviewer_szF5 · 2023-11-07

**Soundness:** 2 fair
**Presentation:** 3 good
**Contribution:** 2 fair
**Rating:** 5
**Confidence:** 3

**Summary:**

This paper proposes a new method for enhancing A3C by introducing state-aware exploration. The method has three components, a ReLu function for advantage estimation, a spectral normalization and dropout. Analysis is provided and experimental results show that the method achieves good performance.

**Strengths:**

The problem in consideration appears interesting and timely.

**Weaknesses:**

- It would be helpful if the authors could improve the motivation of the work. In particular, the Intro and the background do not provide an effective argument as to what problems are really being addressed in this work and why should one care.
- The reviewer would suggest moving the algorithm pseudocode to an earlier place. It is rather inconvenient that Theorem 3.1 is presented before the algorithm.
- Equation (7) seems to assume a Lipschitz condition on v(s)? Please elaborate.
- It might be helpful to explain how many training steps are implemented. Also, would it be possible to show the training curves?

**Questions:**

- It would be helpful if the authors could improve the motivation of the work. In particular, the Intro and the background do not provide an effective argument as to what problems are really being addressed in this work and why should one care.
- The reviewer would suggest moving the algorithm pseudocode to an earlier place. It is rather inconvenient that Theorem 3.1 is presented before the algorithm.
- Equation (7) seems to assume a Lipschitz condition on v(s)? Please elaborate.
- It might be helpful to explain how many training steps are implemented. Also, would it be possible to show the training curves?

---

> ### Author Response · Authors · 2023-11-18
> **Response to Reviewer szF5**
>
> We would like to thank the reivewer for their useful feedback! We are glad the reviewer finds that our work tackles an interesting and timely problem. The reviewer brought up several useful questions and points that we would like to address and shed light on.
>
> ### Questions
>
> > Equation (7) seems to assume a Lipschitz condition on v(s)? Please elaborate.
>
> We hope that the comment above to all reviewers has addressed this question. Please to not hesitate to follow up if you have further clarifying questions concerning this point.
>
> > It might be helpful to explain how many training steps are implemented. Also, would it be possible to show the training curves?
>
> For the MuJoCo-Gym experiments, models interact with each environment for 3 million training steps; for the MuJoCo-Brax experiments, 50 million training steps; and for the ProcGen experiments, 25 million training steps. To save space, we show all training curves in Appendix E. We will make the references to these images in the main text more apparent.
>
> ### Concerns
>
> > The reviewer would suggest moving the algorithm pseudocode to an earlier place. It is rather inconvenient that Theorem 3.1 is presented before the algorithm.
>
> We are moving the algorithm closer to its first reference on Page 4 in the updated manuscript, which we are working to share shortly.

---

> ### Author Response · Authors · 2023-11-22
> **Thank you, again**
>
> Dear Reviewer,
>
> Thank you again for sharing your valuable feedback with us. We strongly believe that the motivation, theory, and empirical results make this submission worthy of acceptance. Moreover, we think our response addresses your concerns and have updated the paper accordingly. Specifically, we added commentary on the Lipschitz condition, included your formatting suggestions, and reworked the abstract and introduction to help communicate the motivation better. If you have no further concerns, we kindly ask you to consider raising your score. Otherwise, we would also be happy to discuss further, should you find the time.
>
> Best regards,
> The Authors

---

### Official Review · Reviewer_QokJ · 2023-11-09

**Soundness:** 3 good
**Presentation:** 2 fair
**Contribution:** 2 fair
**Rating:** 6
**Confidence:** 3

**Summary:**

The paper considers a modified version of A3C algorithm called VSOP by constraining advantage estimates to be positive and applying dropout and spectral normalization both on the actor and the critic networks. Via the application of dropout, the authors tie their presented method to Bayesian inference over critic and actor parameters – such connection for the actor requires that the advantage estimates be gamma distributed. The authors note that since the sign of advantages can be whatever, the gamma requirement is not fulfilled. As a modification, the authors then propose to clip the advantages to only non-negative values and show theoretically that this change corresponds to the policy gradient maximizing a lower bound on the state-value function plus a bounded constant. Motivated by the constant's bound, the authors propose a spectral normalization for the critic weights. VSOP is evaluated Mujoco and ProcGen benchmarks and demonstrates strong performance against several baseline methods.

**Strengths:**

VSOP demonstrates strong performance in multiple environments and is mostly justified, see questions. The presented lower bound view on the policy gradient optimization of clipped advantages is novel and can be used to motivate the choice of spectral normalization for the critic weights.

**Weaknesses:**

Main issues:

While the results presented in the paper are indeed strong, I unfortunately find that the paper is still premature in terms of analysis. Throughout the paper it remained unclear to me how much each added trick contributes to the overall performance. As far as I understand, VSOP was ablated with:
- VSOP (dropout, spectral, relu, thompson)
- A3C (dropout, spectral, thompson)
- No Spectral (dropout, relu, thompson)
- No Thompson (dropout, spectral, relu)

From these ablations it is evident that one cannot determine the contribution of each of the tried tricks. Given the lack of other strong justifications for performance I feel confused by the results. Furthermore, I have some doubts about the theoretical connection between the clipped advantage and MAP estimation, please see questions section.

**Questions:**

* Advantage clipping is motivated by the fact that regular advantages cannot be Gamma distributed, because Gamma has only positive support. While clipping does fix the advantage estimates' support issue, why should this operation make the estimates Gamma distributed as assumed by the theory?

* As authors forthcomingly note, the spectral normalization proves to be detrimental to performance when run in a highly parallelized manner – what could be the reason? Does this also happen with ProcGen environments?

---

> ### Author Response · Authors · 2023-11-18
> **Response to Reviewer QokJ**
>
> We would like to thank the reviewer for their thoughtful review. We are glad that the reviewer finds that "the results presented in the paper are strong" and that the lower bound perpsective for policy gradients is novel. The reviewer mentions some good points that we would like to address.
>
> ### Questions
>
> >Advantage clipping is motivated by the fact that regular advantages cannot be Gamma distributed, because Gamma has only positive support. While clipping does fix the advantage estimates' support issue, why should this operation make the estimates Gamma distributed as assumed by the theory?
>
> The intuition behind assuming a gamma distribution for the clipped advantages is that advantages ideally have zero mean by construction (we subtract the state-action value by its expected state value over actions), so clipping at zero will result in a heavy-tailed distribution. Gamma distributions are sensible hypotheses for heavy-tailed distributions. In the following linked image, for a training run of Humanoid-v4, we plot the marginal histograms for the advantages (left) and the clipped advantages (right) over each training update:
>
> [Histogram of Clipped Advantages](https://i.imgur.com/wExvlMC.png)
>
> The clipped advantage histogram on the right lends evidence to the gamma assumption (at least for the marginal distribution). We may expect multi-modality at the state-action level, which integration over actions may marginalize out at the state level; however, we would still expect a heavy tail in both cases.
>
> > As authors forthcomingly note, the spectral normalization proves to be detrimental to performance when run in a highly parallelized manner – what could be the reason? Does this also happen with ProcGen environments?
>
> We hope the anaysis in the comment to all reviewers helps address these questions. For your convenience we reiterate our thoughts on these specific points here.
>
> In the single-threaded setting, a single agent collects data. This specific experience from a single initialization, coupled with the flexibility of Neural Networks, could result in the objective maximizing a policy that encourages spuriously high-frequency (rather than high-value) value functions when the data is sparse early in training. In this case, regularization from spectral normalization would be beneficial.
>
> Conversely, the algorithm collects data from many agents with unique initializations in the highly parallel setting. Thus, with more diverse and less sparse data, we can expect more robust value function estimates, less likely to be spuriously high-frequency between state transitions. Then, the $K_\pi=1$ assumption induced by spectral normalization may be too strong and lead to over-regularization.
>
> Concerning ProcGen, we report results for an agent trained with spectral normalization under the PPO default settings of 64 threads and 256 time steps per rollout.
>
> We would like to emphasize that our novel theoretical perspective obtains strong results in multiple settings, which we believe is a valuable contribution to the community. Please do not hesitate to follow up if you have further clarifying questions.
>
> ### Concerns
>
> >Throughout the paper it remained unclear to me how much each added trick contributes to the overall performance. As far as I understand, VSOP was ablated with:
>     VSOP (dropout, spectral, relu, thompson)
>     A3C (dropout, spectral, thompson)
>     No Spectral (dropout, relu, thompson)
>     No Thompson (dropout, spectral, relu)
> From these ablations it is evident that one cannot determine the contribution of each of the tried tricks.
>
> Your understanding of the ablation is mostly correct. To help with clarity, we propose to make the following changes to the ablated variant names in Figure 1.:
>
> - RMPG -> VSOP "all-actions"
> - A3C -> VSOP no-ReLU-Adv.
> - No Spectral -> VSOP no-spectral
> - No Thompson -> VSOP no-Thompson
>
> We hyperparameter tune each variant according to the same procedure as for VSOP. The hyperparameter tuning search space includes the dropout rate in the range of 0.0-0.1. As such, we effectivelty ablate dropout rate in the hyperparamter tuning phase to isolate the effect of the added regularization of dropout from the that of Thompson sampling. The estimated optimal dropout rates are 0.025 (VSOP, VSOP no-Thompson, and VSOP no-ReLU), 0.05 (VSOP no-spectral), and 0.0 (VSOP "all-actions").
>
> We respectfully maintain these ablations are sufficient to show that the combination of the four modifications is necessary for the performance of VSOP by demonstrating that the subtraction of any modification results in decreased performance and stability.
>
> We are also interested in the results of seeing the effect size of the remaining nine combinations with respect to A3C, but we have not seen significant evidence that any subset would lead to the performance of VSOP. Lest to justify the substantial computational cost of the hyperparameter tuning and subsequent training runs required for a fair comparison.

---

> > ### Comment · Reviewer_QokJ · 2023-11-22
> > **Raising score**
> >
> > I thank the authors for the detailed response. The added analysis provides more valuable insight to understanding the presented method, which as already stated demonstrates strong performance. Thus I comfortable with raising my score.

---

> > > ### Author Response · Authors · 2023-11-22
> > > **Score Raise**
> > >
> > > We greatly appreciate the score raise and are pleased you found our response and added analysis sufficient!

---

> ### Author Response · Authors · 2023-11-22
> **Thank you, again**
>
> Dear Reviewer,
>
> Thank you again for sharing your valuable feedback with us. We strongly believe that the motivation, theory, and empirical results make this submission worthy of acceptance. Moreover, we think our response addresses your concerns and have updated the paper accordingly. Specifically, we added commentary on the normal-gamma assumption and spectral normalization and strengthened our presentation of the ablation study. If you have no further concerns, we kindly ask you to consider raising your score. Otherwise, we would also be happy to discuss further, should you find the time.
>
> Best regards,
> The Authors

---

### Author Response · Authors · 2023-11-17
**To all reviewers**

We would like to sincerely thank all reviewers for the time and effort that they have put into reviewing our paper! We are very encouraged that there was consensus agreement on the strength of our empirical results and that each reviewer found our approach interesting.

We recognize two common themes in the reviewers' questions. First, each reviewer asks questions about the role of spectral normalization and its relationship to $C(\mathbf{s})$ and K-Lipschitz continuity. Second, the majority of reviewers had questions about the normal-gamma assumption. We address these common concerns in the following two comments. We will post individual responses shortly and are working on incorporating these changes into our manuscript.

---

> ### Author Response · Authors · 2023-11-17
> **Concerning $C(\mathbf{s})$, $K$-Lipschitz continuity, and spectral normalization**
>
> ## Lipschitz Assumption Motivation
> We thank **Reviewer tvnj** for their insightful question on the policy dependence of the Lipschitz constant K. We agree that K would be policy dependent in general and propose to update the notation for the additive term and Lipschitz constant to $C_\pi(\mathbf{s})$ and $K_\pi$, respectively.
>
> In light of this, we believe the roles played by the Lipschitz assumption (to **Reviewer szF5's** question) and the use of critic weight spectral normalization becomes clearer. When we do gradient ascent according to Equation (5), we show that we maximize $$v^*_\pi(\mathbf{s}) \leq v_\pi(\mathbf{s}) + C_{\pi}(\mathbf{s}).$$ We want this optimization to lead to a policy $\pi$ that maximizes value, $v_\pi$, but it could lead to an undesirable policy that instead maximizes $C_{\pi}$.
>
> We show that, $$C_{\pi}(\mathbf{s}) \leq \frac{1}{2}\iint\left|v_\pi(\mathbf{s'}) - v_\pi(\mathbf{s}) \right| dP(\mathbf{s}' \mid \mathbf{S}^t= \mathbf{s}, \mathbf{A}^t= \mathbf{a}) d\Pi(\mathbf{a} \mid \mathbf{S}^t= \mathbf{s}).$$ In theory, a policy that leads to large fluctuations in value, $v_{\pi}$, as the agent transitions from state, $\mathbf{s}$, to state, $\mathbf{s}'$, could maximize this objective.
>
> Assuming that the value function, $v_\pi(\mathbf{s})$, is K-Lipschitz continuous allows us to express this bound as
> $$C_{\pi}(\mathbf{s}) \leq \frac{1}{2}\iint K_{\pi}\left|\left|\mathbf{s'} - \mathbf{s} \right|\right| dP(\mathbf{s}' \mid \mathbf{S}^t = \mathbf{s}, \mathbf{A}^t = \mathbf{a}) d\Pi(\mathbf{a} \mid \mathbf{S}^t = \mathbf{s}),$$
> but this does not solve the problem in itself: it could still be possible to learn a policy that merely maximizes $K_\pi$ instead of $v_\pi(\mathbf{s})$.
>
> ## Spectral Normalization Motivation
> Hence, when we use spectral normalization of the critic weights, we regularize $K$ to be $1$. We find this regularization provides increased performance in most experiments run thus far. But empirically, it does not seem like the pathological behavior of maximizing $C_{\pi}(\mathbf{s})$ is happening to a significant extent even when we do not use spectral normalization. For example, we can see in Figure 1 that the performance of VSOP without spectral normalization is about equal to that of PPO on MuJoCo.
>
> ## Spectral Normalization and Asynchronous Agent Parallelization
> Next, we believe this analysis gives us further insight into understanding **Reviewer QokJ's** question about how we observe spectral normalization detrimental in highly parallel settings.
>
> In the single-threaded setting, a single agent collects data. This specific experience from a single initialization, coupled with the flexibility of Neural Networks, could result in the objective maximizing a policy that encourages spuriously high-frequency (rather than high-value) value functions when the data is sparse early in training. In this case, regularization from spectral normalization would be beneficial.
>
> Conversely, the algorithm collects data from many agents with unique initializations in the highly parallel setting. Thus, with more diverse and less sparse data, we can expect more robust value function estimates, less likely to be spuriously high-frequency between state transitions. Then, the $K_\pi=1$ assumption induced by spectral normalization may be too strong and lead to over-regularization.
>
> Concerning ProcGen, we report results for an agent trained with spectral normalization under the PPO default settings of 64 threads and 256 time steps per rollout.
>
> ## Relaxing the assumption on the $C_\pi(\mathbf{s})$ Bound
> Finally, **Reviewer kpwW** questioned the sensibility of an assumption for the $C_\pi(\mathbf{s})$ bound. Upon a close re-examination of this assumption, we agree it would not generally hold. Fortunately, we can show that it is not necessary and that $C_\pi(\mathbf{s})$ can be smaller than under the zero-expectation assumption, resulting in a smaller additive:
>
> **Lemma.**
>
> $$\text{ReLU}(a) <= |a|$$
>
> *Proof*.
> $$ \text{ReLU}(a) = \max(0, a)$$
> $$\quad\quad\quad\quad= \frac{1}{2}a + \frac{1}{2}|a|$$
> $$\quad\quad\quad\quad= \\{a  \text{ if } a\geq0\text{ else }0\text{ if } a<0$$
> $$\quad\quad\quad\quad\leq \\{a  \text{ if } a\geq0\text{ else }-a\text{ if } a<0 (-a > 0)$$
> $$\quad\quad\quad\quad= |a| $$
>
> Therefore,
>
> $$C_\pi(\mathbf{s}) = \iint \left(v_\pi(\mathbf{s'}) - v^\pi(\mathbf{s}) \right)^+ dP(\mathbf{s}' \mid \mathbf{S}^\mathrm{t} = \mathbf{s}, \mathbf{A}^\mathrm{t} = \mathbf{a}) d\Pi(\mathbf{a} \mid \mathbf{S}^\mathrm{t} = \mathbf{s})$$
> $$\quad\quad\quad\leq \iint \left|v_\pi(\mathbf{s'}) - v_\pi(\mathbf{s}) \right| dP(\mathbf{s}' \mid \mathbf{S}^\mathrm{t} = \mathbf{s}, \mathbf{A}^\mathrm{t} = \mathbf{a}) d\Pi(\mathbf{a} \mid \mathbf{S}^\mathrm{t} = \mathbf{s}) $$
> $$\quad\quad\quad\leq \iint K_\pi\left|\left|\mathbf{s'} - \mathbf{s} \right|\right| dP(\mathbf{s}' \mid \mathbf{S}^t = \mathbf{s}, \mathbf{A}^\mathrm{t} = \mathbf{a}) d\Pi(\mathbf{a} \mid \mathbf{S}^\mathrm{t} = \mathbf{s})$$

---

> ### Author Response · Authors · 2023-11-17
> **Concerning the Normal-Gamma Assumption**
>
> ## Is the normal-gamma assumption necessary?
>
> The gamma-normal assumption allows us to interpret adding dropout and weight-decay regularization as sensible approximate Bayesian inference without adding complex computational overhead to the original A3C optimization algorithm. As such, this assumption primarily serves to ground Thompson sampling through approximate Bayesian inference and is not requisite for Theorem 1. As with the original result of the policy gradient theorem, the results in Equations (5-6) do not make any distributional assumptions on $\pi$ and should hold for all policies with differentiable probability densities/distributions.
>
> ## Are the clipped advantages gamma distributed?
>
> The intuition behind assuming a gamma distribution for the clipped advantages is that advantages ideally have zero mean by construction (we subtract the state-action value by its expected state value over actions), so clipping at zero will result in a heavy-tailed distribution. Gamma distributions are sensible hypotheses for heavy-tailed distributions. In the following linked image, for a training run of Humanoid-v4, we plot the marginal histograms for the advantages (left) and the clipped advantages (right) over each training update:
>
> [Histogram of Clipped Advantages](https://i.imgur.com/wExvlMC.png)
>
> The clipped advantage histogram on the right lends evidence to the gamma assumption (at least for the marginal distribution). We may expect multi-modality at the state-action level, which integration over actions may marginalize out at the state level; however, we would still expect a heavy tail in both cases.

---

### Author Response · Authors · 2023-11-20
**Paper and Appendices Update**

We are thankful to the reviewers for their patience. We have finished updating the paper and believe that the resulting changes have addressed the most pressing of their concerns. Again, we are thankful for the reviewer's comments and insights, which, we believe, have contributed to a significantly improved presentation of our work. We summarize the substantive changes below and look forward to any further feedback.

### Summary of Changes

- **$C_\pi(\mathbf{s})$ Bound (all reviewers).** In the main text, we have made dependency on $\pi$ more explicit and dropped the unnecessary assumption. In the appendices, we have Added lemma C.4 and the above commentary as Appendix D.1.

- **Normal-gamma assumption (QokJ, tvnj, kpwW).** We have added the above commentary as Appendix D.2.

- **Ablation of Mechanisms (QokJ, kpwW).** We renamed ablated VSOP variants (VSOP, no-spectral, no-Thompson, all-actions, no-ReLU Adv.), updated results with hyperparameter tuned variants, and we have added probability of improvement metrics and sample efficiency plots.

- **Thompson sampling (kpwW).** We have added our response to Reviewer kpwW's questions about Thompson sampling and Bayesian inference as Appendix D.3.

- **Motivation (szF5).** We have reworked the abstract and introduction to be more in line with the structure of the paper. We hope this serves to clarify the motivation and contributions of our work.

- **Description of Metrics (kpwW).** We have updated the description of metrics in Section 5 as: "We use the rliable package to evaluate robust normalized median (Median), interquartile mean (IQM), mean (Mean), optimality gap (OG), and probability of improvement (Prob. Improve) metrics."

- **Typos (kpwW).** We have corrected the Normal-Gamma distribution formula in Equation (4) and added the missing $\nabla$ in Appendix C.1.

- **Algorithm 1 (szF5).** We have moved the algorithm pseudocode from page 5 to page 4.

---

### Meta-Review · Area_Chair_e6PY · 2023-12-04

**Metareview:**

This paper presents a novel method for enhancing the effectiveness of the Asynchronous Advantage Actor-Critic (A3C) algorithm by incorporating state-aware exploration.

**Reviewers have reported the following strengths:**

- Importance of the studied topic;
- Strong empirical performance.

**Reviewers have reported the following weaknesses:**

- Quality of writing;
- Motivation for the proposed method;
- Unclear contribution.

**Decision**

The authors' rebuttal helped solve the Reviewers' doubts, and some of them have increased the score. However, concerns remained about the motivation and novelty of the method. This paper seems close to being in good shape for acceptance, but improvement in the presentation and motivation is needed. I encourage the authors to address the remaining Reviewers' concerns in a future submission.

**Justification For Why Not Higher Score:**

N/A

**Justification For Why Not Lower Score:**

N/A

---

### Decision · Program_Chairs · 2024-01-16

Reject